# Effects of a Short Daytime Nap on the Cognitive Performance: A Systematic Review and Meta-Analysis

**DOI:** 10.3390/ijerph181910212

**Published:** 2021-09-28

**Authors:** Frédéric Dutheil, Benjamin Danini, Reza Bagheri, Maria Livia Fantini, Bruno Pereira, Farès Moustafa, Marion Trousselard, Valentin Navel

**Affiliations:** 1CNRS, LaPSCo, Physiological and Psychosocial Stress, University Hospital of Clermont-Ferrand, CHU Clermont-Ferrand, Occupational and Environmental Medicine, WittyFit, F-63000 Clermont-Ferrand, France; 2Preventive and Occupational Medicine, University Hospital of Clermont-Ferrand, F-63000 Clermont-Ferrand, France; benjamin.danini@gmail.com; 3Department of Exercise Physiology, University of Isfahan, Isfahan 8174673441, Iran; will.fivb@yahoo.com; 4NPsy-Sydo, Sleep Disorders, University Hospital of Clermont-Ferrand, CHU Clermont-Ferrand, F-63000 Clermont-Ferrand, France; maria_livia.fantini@uca.fr; 5Clinical Research and Innovation Direction, CHU Clermont-Ferrand, F-63000 Clermont-Ferrand, France; bpereira@chu-clermontferrand.fr; 6Emergency Department, CHU Clermont-Ferrand, F-63000 Clermont-Ferrand, France; fmoustafa@chu-clermontferrand.fr; 7Neurophysiology of Stress, Armies’ Biomedical Research Institute, Armies’ Health Service, F-91220 Brétigny sur Orge, France; marion.trousselard@gmail.com; 8CNRS, INSERM, GReD, University Hospital of Clermont-Ferrand, Ophthalmology, CHU Clermont-Ferrand, Université Clermont Auvergne, F-63000 Clermont-Ferrand, France; valentin.navel@hotmail.fr

**Keywords:** daytime nap, cognitive performance, work, prevention

## Abstract

Background: Napping in the workplace is under debate, with interesting results on work efficiency and well-being of workers. In this systematic review and meta-analysis, we aimed to assess the benefits of a short daytime nap on cognitive performance. Methods: PubMed, Cochrane Library, ScienceDirect and PsycInfo databases were searched until 19 August 2021. Cognitive performance in working-aged adults, both before and following a daytime nap or under control conditions (no nap), was analysed by time and by type of cognitive function (alertness, executive function and memory). Results: We included 11 studies (all in laboratory conditions including one with a subgroup in working conditions) for a total of 381 participants. Mean duration of nap was 55.4 ± 29.4 min. Overall cognitive performance did not differ at baseline (t0) between groups (effect size −0.03, 95% CI −0.14 to 0.07), and improved in the nap group following the nap (t1) (0.18, 0.09 to 0.27), especially for alertness (0.29, 0.10 to 0.48). Sensitivity analyses gave similar results comparing only randomized controlled trials, and after exclusion of outliers. Whatever the model used, performance mainly improved until 120 min after nap, with conflicting results during the sleep inertia period. Early naps in the afternoon (before 1.00 p.m.) gave better cognitive performance (0.24, −0.07 to 0.34). The benefits of napping were independent of sex and age. Duration of nap and time between nap and t1 did not influence cognitive performance. Conclusions: Despite the fact that our meta-analyses included almost exclusively laboratory studies, daytime napping in the afternoon improved cognitive performance with beneficial effects of early nap. More studies in real work condition are warranted before implementing daytime napping at work as a preventive measure to improve work efficiency.

## 1. Introduction

Napping in the workplace is under consideration, with putative benefits on work efficiency and well-being of workers [1]. Interestingly, productivity at work does not increase with working time [2], with a lack of sleep having a strong negative impact on work productivity [3]. Even if several countries benefited from the virtues of napping for millennia [4], rest time at work is still perceived as a waste of time [5,6] or a sign of laziness [7,8] in western countries. For example, the right to nap has been enshrined in the Chinese Constitution since 1949 [8,9]. In Japan, napping is ideologically accepted in the world of work it is even highly recommended by some companies. It is called “inemuri”, which literally means “to be asleep while present” [10]. Many scientists recommend taking an afternoon nap to increase alertness, specifically after lunch [11], to stimulate creativity, to strengthen our memory [12,13,14] and to improve the performance of complex tasks (executive function) [15]. The putative detrimental effects of napping at awakening, during the sleep inertia period, lack of data in real work conditions [16]. Research has demonstrated that a short nap can benefit cognitive performance the most [17,18,19], and the recuperative value of a nap is also dependent on when the nap is taken within the day [20]. World-famous companies, such as Google, NASA, HuffPost or Samsung, provide to their employees rest areas at work or dedicated nap furniture [21,22]. Considering the numerous studies on the effects of napping at work, a meta-analysis would allow us to synthesize all available evidence-based data from the literature. To the best of our knowledge, no meta-analysis assessed the effects of a daytime nap on cognitive performance. Therefore, we hypothesized that (1) napping at work during day time can benefit cognitive performance, (2) all types of cognitive performance (alertness, memory, executive function) can be improved by a nap, (3) duration of benefits may be prolonged with few detrimental effects during sleep inertia, (4) these effects may also be linked with characteristics of napping (duration, time between nap and test, time of the day) and individuals (age, gender), and conditions of realization of studies (real work or laboratory). Thus, we aimed to conduct a systematic review and meta-analysis on the effects of a short daytime nap on cognitive performance.

## 2. Methods

### 2.1. Literature Search

All the studies reporting the effects of a daytime nap on cognitive performance compared to a control group without nap were reviewed. The PubMed, Cochrane Library, ScienceDirect and PsycInfo databases were searched until 19 August 2021, with the following keywords: “daytime nap” OR “napping” AND “work” OR “occupation”. Animal studies were excluded. The search was not limited to specific years and no language restrictions were applied. The search strategy included working-age adults (more than 18 years old), working only on daytime hours. Studies related to night shift work were excluded, as studies mentioning the use of therapeutic adjuvant in addition to nap. Studies related to the effect of nap on fatigue were also excluded because we aimed to analyze only cognitive performance. To be included, articles needed to be controlled studies describing our primary outcome, i.e., the assessment of cognitive performance after a short daytime nap. The search strategy is presented in Figure 1. Two authors (BD and LF) conducted all the literature searches and collated the abstracts. Three authors (BD, LF and VN) separately reviewed the abstracts and decided the suitability of the articles for inclusion based on the selection criteria. A fourth author (FD) was asked to review the articles where consensus on suitability was debated. All authors then reviewed the eligible articles. We followed the PRISMA guidelines (Appendix A).

### 2.2. Quality of Assessment

The “Consolidated Standards of Reporting Trials” (CONSORT) [23] and the “STrengthening the Reporting of OBservational studies in Epidemiology” (STROBE) [24] statements were, respectively, used for checking the quality of randomised and non-randomised controlled study reporting. One point was attributed per item or sub-item, with a maximal score of 37 and 32, respectively, then converted into a percentage (Appendix A). The quality of included studies was also assessed using the “Scottish Intercollegiate Guidelines Network” (SIGN) [25] Methodology checklist. Items assessing internal validity were quoted as “Yes”, “No”, or “Can’t say”. Overall assessment of studies was quoted as “unacceptable”, “acceptable” or of “high quality”. One point was attributed per response “Yes” or “High Quality”, for a maximal score of 10 for randomised controlled trials (RCT) and 7 for non-RCT, then converted into a percentage (Figure 2).

### 2.3. Statistical Considerations

Statistical analysis was conducted using Stata software (v15, StataCorp, College Station, TX, USA) [26,27,28,29,30,31,32,33]. Baseline characteristics were summarized for each study sample and reported as mean (standard deviation) and number (%) for continuous and categorical variables, respectively. Heterogeneity in the study results was evaluated by examining forest plots, confidence intervals (CI) and using formal tests for homogeneity based on the I-squared (I^2^) statistic. I^2^ is easily interpretable and the most common metric to measure the magnitude of between-study heterogeneity. I^2^ values range between 0% and 100% and are typically considered low for <25%, modest for 25–50%, and high for >50%. This statistical method generally assumes heterogeneity when the *p*-value of the I^2^ test is <0.05. For example, a significant heterogeneity may be due to the variability between the characteristics of the studies such as those of workers (age, sex, etc.), or those of nap (duration, time of the day). Random effects meta-analyses (DerSimonian and Laird approach) were conducted when data could be pooled [34]. *p* values less than 0.05 were considered statistically significant. We conducted a meta-analysis on the effects of daytime napping on cognitive performance at work. We first conducted one meta-analysis on cognitive performance at baseline (t0) to verify that groups were comparable before the intervention (nap). Then we conducted a meta-analysis on overall cognitive performance after a nap (t1) between groups (nap vs. no nap), stratified on the type of cognitive functions (alertness, executive functions, and memory) and on the time of the analysis: <30 min after the nap, 31 min to 60, 61 to 120 min, and >121 min. Then, we conducted two separate meta-analyses on the effects of nap versus baseline (t1 vs. t0): one within the nap group, and one within the control group. Finally, we conducted a meta-analysis on changes in performance between groups ((t1 − t0)/t0). We also stratified those meta-analyses on the type of cognitive functions and the time between nap and tests. We described our results by calculating the effect size (ES, standardized mean differences—SMD) of cognitive performance for each dependent variable [34]. A positive ES denoted improved performance. All meta-analyses were also conducted after the exclusion of non-RCT. For rigor, funnel plots (metafunnels) of these meta-analyses were used to search for potential publication bias. In order to verify the strength of the results, further meta-analyses were then conducted excluding outliers, i.e., studies that were not evenly distributed around the base of the funnel [35]. When possible (sufficient sample size), meta-regressions were proposed to study the effects of daytime nap on cognitive performance relevant parameters such as the workers’ gender, age, the time of nap beginning, duration of nap, time between test and nap. Results were expressed as regression coefficients and 95%CI.

## 3. Results

An initial search on the selected databases found 1002 articles. Once the duplicate articles were removed and using the inclusion criteria, this number was reduced to 17 articles dealing with cognitive performance after a daytime nap, which was included in the systematic review. After a detailed analysis of the articles, three [36,37,38,39] were excluded because the numerical data were not available to perform a statistical analysis of the results. Another three articles [13,39,40] were also excluded due to a lack of baseline (t0) in the protocol which distorts comparability of tests after the nap, and one article [39] was excluded because of the two previous criteria. The final sample included in our meta-analysis, therefore, reports 11 articles [6,41,42,43,44,45,46,47,48,49,50] (Figure 1). All articles are written in English. The characteristics of included studies were available in Appendix A.

### 3.1. Quality of Articles

The assessment of the quality of the 11 included studies that were included was performed using the CONSORT for the six RCT [6,43,44,45,46,50] and STROBE criteria for the five non-RCT [41,42,47,48,49], with, respectively, a mean score of 42.9 ± 8.5%, ranging from 31.4 [45] to 57.1% [46] and a mean score of 41.9 ± 6.5%, ranging from 34.4 [48] to 50% [41] (Appendix A). Using the SIGN criteria, RCT had a mean score of 61.7 ± 9.8%, ranging from 50.0 [44] to 80.0% [46], and all non-RCT had a score of 83.3% [41,42,47,48,49]. Thus, the total score using SIGN for the 11 studies was 71.5 ± 13.27%, ranging from 50 [44] to 83.3% [41,42,47,48,49] (Figure 2). Overall, the studies performed best in the Section 1 and worst in the title/abstract and Section 3. Most studies (9/11, i.e., 81.8%) mentioned ethical approval [6,41,42,43,44,45,46,47,50].

### 3.2. Inclusion Criteria of Participants

The main inclusion criteria were to be an adult [6,42,43,44,45,46,47] or a student [41,48,49,50], in good health [6,42,43,44,45,46,48,50] or to be a good sleeper [46]. To be habitual afternoon nappers was necessary for one study [43]. The main exclusion criteria were night shift workers [41,44,50], any sleep disorders [6,42,44,45,46], smokers [43,45,46], caffeine and alcohol consumption [6,42,43,44,45,46,48,49], or excessive physical exercise the day of the study [42,43], psychoactive drugs or medication use [43,45,46,49,50], medication that might alter sleep architecture or ability to fall asleep [6,44,48], regular nappers [45], excessive morning-evening type [45], a short sleep duration per night [50] or anxiety and depression symptoms [50].

### 3.3. Population

#### 3.3.1. Sample Size

381 participants were included: 209 in the nap group and 188 in the control group, among which 16 were enrolled in a crossover condition. Mean population size was 34.6 ± 20.0 ranging from 8 [42,47] to 76 [50]. Mean proportion of subjects in the nap group was 51.7 ± 7.9% ranging from 41.4 [48] to 66.7% [44].

#### 3.3.2. Sex

Proportion of male in the nap group was 55.5 ± 10.9%, ranging from 45.2 [44] to 75% [42], and 46.7 ± 20.1% in the control group, varying from 14.3 [44] to 75% [42]. Five studies [43,45,46,47,49] did not report proportion of male.

#### 3.3.3. Age

Mean age in the nap group was 33.7 ± 2.6 years and 29.6 ± 1.8 years in the control group, ranging from 20.1 [48] to 70.4 years [44] and from 20.1 [48] to 73.9 years [44], respectively.

#### 3.3.4. Other

Other characteristics were seldom reported, such as body mass index which was reported only in one study [41] and smoking in three studies [43,45,46]. The lack of data for those variables precluded further analysis.

### 3.4. Aims and Outcomes of Included Articles

The primary objective of most studies (10/11, i.e., 90.9%) was to examine the effect of a short midday nap on cognitive performance [6,41,43,44,45,46,47,48,49,50]. The aim of one study was to investigate the effect of sleep inertia after a daytime nap [42].

### 3.5. Study Designs

All included studies were interventional prospective controlled trials, among two studies (18.2%) with a crossover design [42,47] and six were RCT (54.5%) [6,43,44,45,46,50].

### 3.6. Characteristics of Intervention

Duration of nap: Mean duration of nap, i.e., time of sleep duration was 55.4 ± 29.4 min, varying from 15 [47] to 90 min [6,44,45,50].

Number of naps per day: All studies evaluated the effect of one nap per day in the early afternoon [6,41,42,43,44,45,46,47,48,49,50].

Time of nap beginning: All studies gave information on the time of the nap. Mean hour time of nap was 1.32 pm (SD = 44.4 min), varying from 12.30 pm [47] to 14.45 pm [43].

The study environment took place in a sleep laboratory in all studies [6,41,42,43,44,45,46,47,48,49,50]. Only one study was also in the usual work environment, i.e., in medical departments of first-year internal medicine residents [41].

The activity of control group during nap: During the nap of the intervention group, the control group had the opportunity to rest [41,42,44,45,48,50] or to do a quiet activity like watching TV [6,43,47], reading [46] or playing a card game [45].

Time of testing: All studies assessed baseline (t0) cognitive performance before the nap [6,41,42,43,44,45,46,47,48,49,50]. The mean time between baseline testing and the intervention, nap or no-nap, was 86.3 ± 80.4 min, varying from 10 [47] to 285 min [41]. Mean time between nap and the repeat test (t1) was 77.0 ± 59.9 min, varying from 1.5 [42] to 240 min [48].

### 3.7. Measurements of Cognitive Performance

Cognitive performance was evaluated with measures of alertness in five studies [41,42,43,46,47], executive functions in six studies [41,42,43,48,49,50] and memory in seven studies [6,43,44,45,48,49,50] (details of test used for measures of cognitive function in Appendix A). Alertness was evaluated with seven tests: Electroencephalographic and electrooculographic recording [41], Conner’s Continuous Performance Test Version 5 [41], Arrow-orientation task [42], Test of Attentional Performance [43], Ball and cup task [46], Choice reaction time task [47] and an Alertness scale [47]. Executive functions were evaluated with five tests: Conner’s Continuous Performance Test Version 5 [41], Arrow-orientation task [42], Auditory oddball task [46], Mirror Tracing Task [43,48,50], and Maze learning task [49]. Memory was evaluated with nine tests: Paired associates learning [43], Digit span backwards task [43,48], Verbal learning and memory test [44], Motor adaptation task [44], Sequential finger-tapping task [45], Direct associative (face–object) memory [6], Paired associates task [48], Semantically unrelated paired associates [49], and Word-pair task [50].

### 3.8. Meta-Analysis on Performance between Groups at Baseline (t0) by Activity

Overall cognitive performance at baseline (t0) did not differ between nap vs. control groups (effect size −0.03, 95% CI −0.14 to 0.07, I^2^ = 21.2%), nor after exclusion of outliers [41,44,50] (−0.01, −0.10 to 0.09, I^2^ = 0.0%), nor after exclusion of non-RCT [41,42,47,48,49] (−0.04, −0.16 to 0.09, I^2^ = 17.2%) (Appendix A). Stratification by cognitive functions (memory, alertness, and executive functions) demonstrated similar results, i.e., no difference between nap and control groups for the overall analyses, as well as after exclusion of outliers [41,44,50] and after exclusion of non-RCT [41,42,47,48,49]—except for executive functions that were lower in the nap group compared to the control group but only in the sensitivity analyses with two RCT (Appendix A).

### 3.9. Meta-Analysis on Overall Effects of Nap between Groups

Overall cognitive performance improved in the nap group following the nap (t1) compared to the control group (effect size 0.18, 95% CI 0.09 to 0.27, I^2^ = 4.3%) (Figure 3, Figure 4, Appendix A). Stratifying analysis by type of cognitive functions, all types of cognitive performance increased or tended to increase after napping compared with controls (memory: 0.11, −0.01 to 0.24, I^2^ = 0.0%; alertness: 0.29, 0.10 to 0.48, I^2^ = 38.9%; and executive functions: 0.23, 0.00 to 0.47, I^2^ = 0.0%). Stratifying analysis by time of analysis, cognitive performance improved less than 30 min (0.22, 0.03 to 0.42, I^2^ = 33.6%), 61 to 120 min (0.28, 0.09 to 0.48, I^2^ = 1.0%) and more than 120 min after nap (0.20, 0.04 to 0.37, I^2^ = 0.0%). Similar results were found after exclusion of outliers [47,50] and after exclusion of non-RCT [41,42,47,48,49], except less than 30 min after nap which was no more significant after exclusion of outliers (Figure 3, Figure 4, Appendix A).

### 3.10. Sensitivity Analysis

Meta-analysis on effects of nap within the nap group (Figure 5, Appendix A) and meta-analysis on performance change (Figure 5 and Appendix A) between groups demonstrated similar overall effects of napping (effect size 0.34, 95% CI 0.23 to 0.45, I^2^ = 35.8%, and 1.26, 0.79 to 1.73, I^2^ = 93.3%, respectively). Similar results were demonstrated stratifying by type of cognitive performance and by time, globally, after exclusion of outliers [41,44,45,47,50] and of non-RCT [41,42,47,48,49] (Figure 5, Appendix A). Lastly, comparisons of performance at t1 vs. t0 within the control group (Figure 5, Appendix A) mostly did not show differences for overall cognitive performance (0.08, −0.03 to 0.19, I^2^ = 31.8%), as well as stratifying the analysis by type of cognitive functions and by time, globally, after exclusion of outliers [41,44,45,47,50] and of non-RCT [41,42,47,48,49] (Figure 5, Appendix A). Detail text is available in Appendix A.

### 3.11. Metaregressions

Men tended to have poorer cognitive performance than women at baseline (coefficient −0.02, 95% CI −0.03 to 0.002, *p* = 0.086), as well as older participants compared to younger ones (0.01, 0.01 to 0.02, *p* < 0.001). The benefits of napping on changes in performance were independent of sex and age (*p* > 0.05). Napping early at the beginning of the afternoon (before 1 p.m.) was more effective on cognitive performance compared with after 1 p.m. (0.28, 0.10 to 0.46, *p* = 0.003). Duration of nap and time between nap and t1 did not influence cognitive performance (Figure 6).

## 4. Discussion

The main findings were that napping in the afternoon improved cognitive performance, especially for alertness. However, the duration of benefits should warrant further studies, as napping seemed to improve performance within two hours, with conflicting results during the sleep inertia period. An early nap in the afternoon may be beneficial to cognitive performance. Gender and age did not influence cognitive performance, as well as the duration of nap and time between nap and t1. Our meta-analyses included almost exclusively laboratory studies so results were more difficult to transpose in real work conditions.

### 4.1. Nap and Cognitive Performance

Many people take daytime naps, with the frequency of napping varying considerably depending on the country, from 36% to 80% [51]. Reasons for napping are multiple: in response to sleep loss (i.e., replacement napping), in preparation for sleep loss (i.e., prophylactic napping), or just for enjoyment (i.e., appetitive napping) [52]. Our study demonstrated that napping in the afternoon improved all types of cognitive performance. Napping is particularly beneficial to performance on tasks [53], such as addition, logical reasoning, reaction time [54], and symbol recognition [55]. Napping appears beneficial for all types of memory, either procedural [56], declarative [57] or short-term memory [19]. Daytime napping offers various other benefits such as relaxation, reduced fatigue [58] and improve mood [55]. Napping can boost creativity [59,60] and productivity [61], improve physical performance [62] and help people to cope with fatigue related to shiftwork [63,64,65]. Daytime sleep may also offer cardiovascular benefits in the form of greater cardiovascular recovery from psychological stress [66]. For example, taking a midday nap, occasionally or at least three times per week, was reported to be inversely associated with coronary mortality. This association was particularly evident among working men [67].

### 4.2. Duration of Benefits and Sleep Inertia

The literature reports that the benefits of daytime napping may last 2.5 h [68], with conflicting results during the sleep inertia period, i.e., after awakening [55,69]. These findings were in accordance with results from our study, in which the positive effects of the nap were mainly 30–120 min following awakening. For the 30 min after napping, results were mitigated and variable, depending on sensitivity analyses, reflecting putative effects of sleep inertia [16]. Even if we did not find an influence on the duration of the nap, the literature suggests that short naps may benefit more on cognitive performance—possibly because napping more than 30 min produces sleep inertia, making nap benefits obvious only after a delay [55]. Sleep inertia reflects a transition from a sleep state to a waking state [19] and is characterized by a reduction in the ability to think and perform upon awakening due to sleep [55]. This period is a state of grogginess, confusion [70] and lowered arousal [71,72]. The magnitude of sleep inertia is mostly dependent on the quantity of slow-wave sleep contained within the nap [73]. Sleep inertia is greater following longer naps that typically contain more slow-wave activity than shorter naps [11,74,75]. So, to avoid sleep inertia, naps should be short (20–30 min), and should not occur at the bottom of the circadian phase [76,77]. Paradoxically, in older adults [78] and not in middle-aged workers, napping might both increase morbidity [79,80,81,82,83]—cardiovascular disease, falls and cognitive impairment—and mortality [84,85,86,87]. Daytime napping could also diminish the quality of sleep at night [8].

### 4.3. Environmental and Individual Characteristics

The recuperative value of a nap depends also on the 24-h circadian rhythm. We showed that early nap in the afternoon has greater benefits, in line with the literature [88,89]. Our organism is physiologically programmed to rest in the afternoon. Our biological clock controls a biphasic rhythm with two periods conducive to sleep, varying with body temperature. The first peak of drowsiness occurs between 1 and 5 a.m., the second twelve hours later, i.e., between 1 to 5 p.m. [90]. The decrease in alertness within the afternoon is wrongly associated with digestion [91], but is mainly due to our circadian rhythm [92,93]. So it could suggest a night sleep and a nap in the early afternoon, i.e., when we are naturally less vigilant [90]. However, ideally, workers may also benefit to nap according to their circadian rhythm rather than clock time. Other factors such as individual characteristics (e.g., age, gender) may also influence the benefits of napping [55]. Elderly nap more frequently than youngers [55,94,95,96]. Many factors are likely to contribute, such as disturbed night-time sleep [97,98], age-related phase advance of circadian rhythm [99], comorbidities [96] or some combination of those [57]. In our study, gender and age did not influence cognitive performance after a nap. However, literature suggested sex differences in benefits of daytime sleep, for example, greater benefits on memory for women [100]. A lab environment does not completely duplicate the real world. As our meta-analyses included mainly lab studies, the applicability of our results to real work must be inferred [101].

### 4.4. Limitations

Our study has some limitations like all meta-analyses [102]. Meta-analyses inherit the limitations of the individual studies of which they are composed and therefore are subjected to the bias of included studies. We conducted the meta-analyses on published articles only so there are potentially exposed to the publication bias [103,104]. Studies with positive results are more likely to be published than studies with negative results, which may also lead to this bias. Moreover, we included a limited number of participants. Our results were based on studies with small sample sizes, that typically have low statistical power, and large standard errors [105]. As a result, a meta-analysis based on these studies can produce effect sizes that are very heterogeneous, and the findings may lead to erroneous inferences [102]. However, this risk is limited because funnel plots had homogenous distributions [106]. We limited the influence of extreme results and heterogeneity by reperforming analyses after exclusion of non-randomized studies and of those with results not evenly distributed around the funnel plots. Though there were similarities between the inclusion criteria, they were not identical. Another limitation of our meta-analysis is the quality of the control group. Participants from the control group did not take a nap but realized relaxing activities which are very different from real work. This could minimize our results. Nearly all data were also in laboratory condition, except one study. Consequently, it is hard to make conclusions of the effects of napping on cognitive performance during daytime work, decreasing the external validity [107] and strongly enhancing the level of evidence for napping at work. The environment differs widely between working conditions and sleep lab rooms in hospitals. Consequently, the final number of patients included in the meta-analysis was not very high and may preclude generalizability.

## 5. Conclusions

Napping in the afternoon improved cognitive performance and especially alertness, until two hours after the nap, with conflicting results during the sleep inertia period. Early nap in the afternoon was more effective on cognitive performance. However, our meta-analyses included almost exclusively laboratory studies. Before recommending daytime napping at work as a preventive strategy, further studies should evaluate the effects of naps on cognitive performance in real work conditions to make the results more generalizable.

## Figures and Tables

**Figure 1 ijerph-18-10212-f001:**
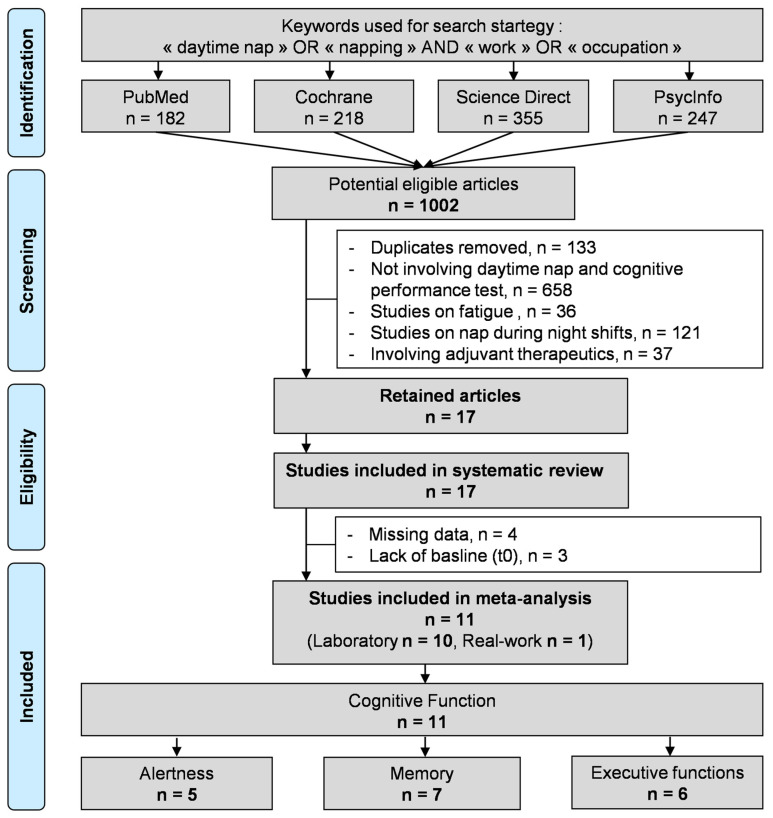
Search strategy.

**Figure 2 ijerph-18-10212-f002:**
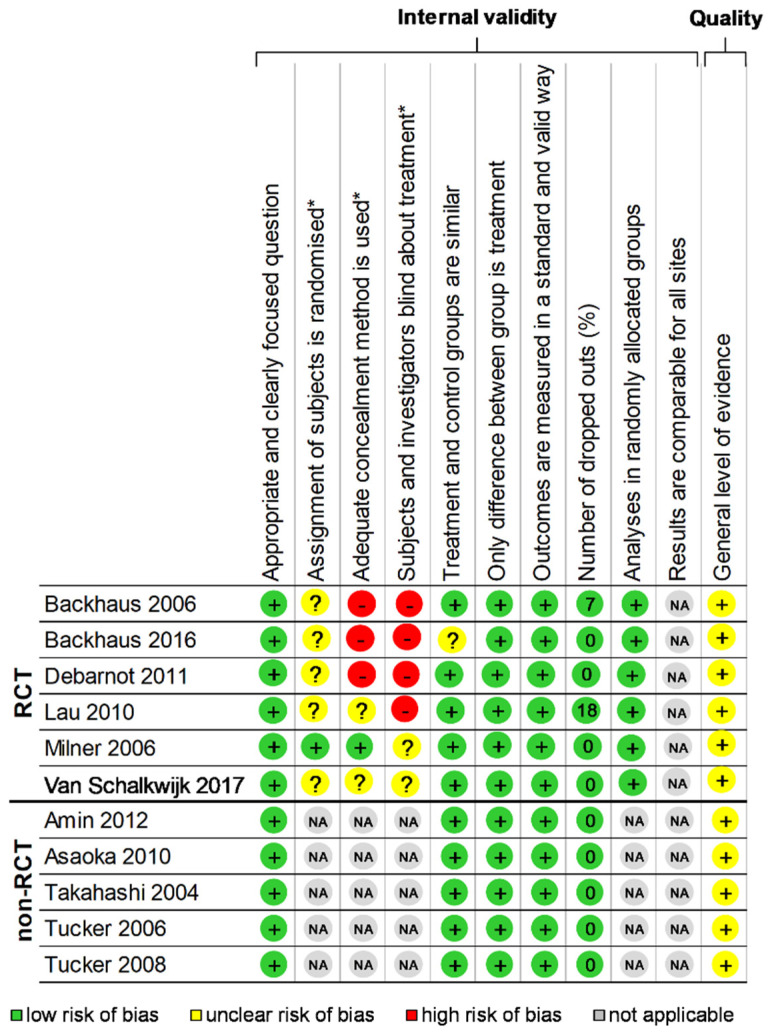
Methodological quality of included articles and summary bias risk. Using the “Scottish Intercollegiate Guidelines Network” (SIGN) Methodology checklist 2 Yes: +; No: − Can’t say: ?; Not applicable: NA; RCT: Randomized controlled trials; * item only for randomized studies.

**Figure 3 ijerph-18-10212-f003:**
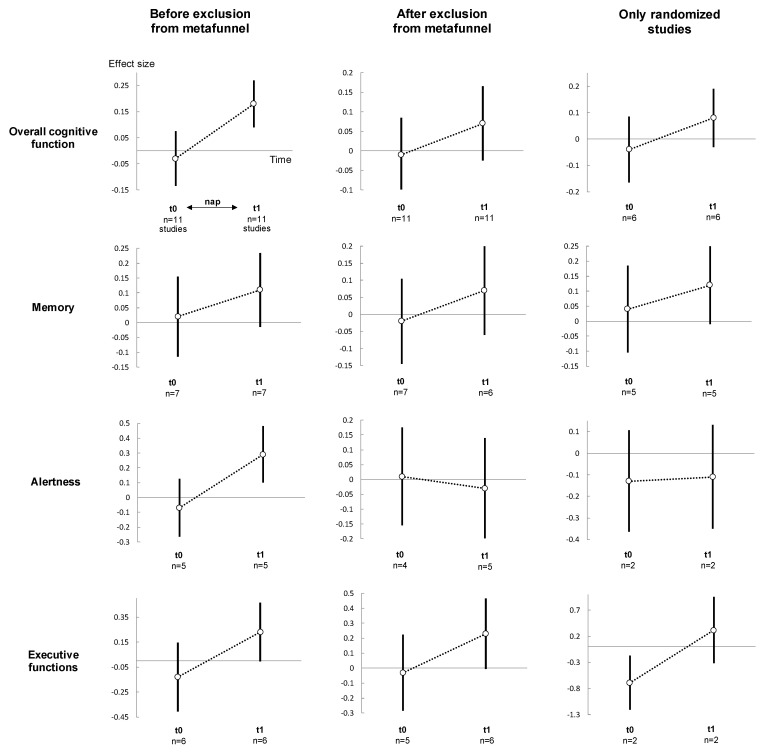
Summary of meta-analysis on cognitive performance between groups (nap vs. control), for each cognitive function: before (**left**), after (**middle**) exclusion of studies not evenly distributed around the funnel plot and (**right**) non-randomised controlled trials. For details of the meta-analysis at each analysis time, please see Appendix A for meta-analyses t1. For details of the meta-analysis on each cognitive function, please see Appendix A for meta-analyses at t0, and Figure 4 for t1. For details of the meta-analysis on each group, please see Figure 4 and Appendix A for the nap group, Figure 5 for the control group. t0: baseline; t1: after intervention (nap or no-nap). Each overall summary of a meta-analysis is represented in the graph by a dot on a vertical line. The black dots represent the overall pooled-effect estimate of individual meta-analyses (pooled effect size—ES), and the length of each vertical line around the dots represents their 95% confidence interval (95CI). Shorter lines represent a narrower 95CI thus higher precision around pooled-ES. Conversely, longer lines represent a wider 95CI and less precision around pooled-ES. The black solid horizontal line represents the null estimate (with a value of 0 for pooled-ES). Vertical lines that cross the null horizontal line represent a non-significant overall summary of the meta-analysis at each time analysed (t0 and t1).

**Figure 4 ijerph-18-10212-f004:**
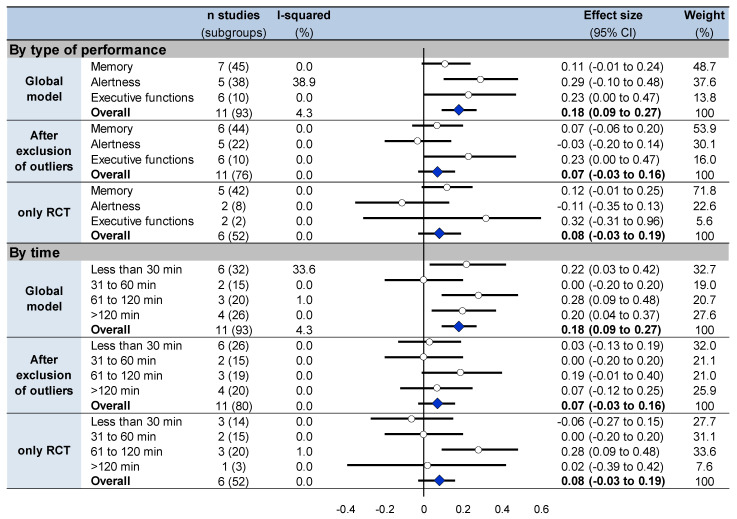
Summary of meta-analyses on cognitive performance stratified on type of cognitive function and on time of analysis at t1 between groups. Each summary of meta-analysis is presented in three conditions: global model with all the studies, after exclusion of outliers (studies not evenly distributed around the funnel plot) and with only randomized controlled trials. Each summary of several meta-analyses is represented in the forest plot by a dot on a horizontal line. The black dots represent the pooled-effect estimate (pooled effect size—ES), and the length of each line around the dots represents their 95% confidence interval (95CI). Shorter lines represent a narrower 95CI thus higher precision around pooled-ES. Conversely, longer lines represent a wider 95CI and less precision around pooled-ES. An overall summary of the results of the meta-analyses pooled-estimate (result of the overall meta-analysis) is represented by a blue lozenge at the end of the graph. The black solid vertical line represents the null estimate (with a value of 0 for pooled-ES). Horizontal lines that cross the null vertical line represent the non-significant overall summary of the meta-analysis. Bold numbers represent the overall result of each meta-analysis.

**Figure 5 ijerph-18-10212-f005:**
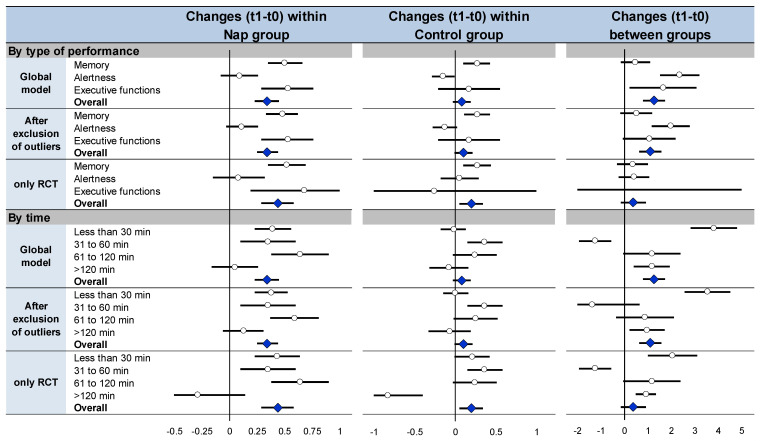
Summary of sensitivity analysis: meta-analyses on cognitive performance at t1 compared with baseline (t0) within the nap group (**left**), meta-analyses on cognitive performance at t1 compared with baseline (t0) within the control group (**middle**), meta-analyses on changes in performance between t1 and t0 ((t1 − t0)/t0) between nap and control groups (**right**) Each summary of meta-analysis is presented in three conditions: global model with all the studies, after exclusion of outliers (studies not evenly distributed around the funnel plot) and with only randomized controlled trials For details of each meta-analysis, please see Appendix A. Each summary of several meta-analyses is represented in the forest-plot by a dot on a horizontal line. The black dots represent the pooled-effect estimate (pooled effect size—ES), and the length of each line around the dots represents their 95% confidence interval (95CI). Shorter lines represent a narrower 95CI thus higher precision around pooled-ES. Conversely, longer lines represent a wider 95CI and less precision around pooled-ES. An overall summary of the results of the meta-analyses pooled-estimate (result of the overall meta-analysis) is represented by a blue lozenge at the end of the graph. The black solid vertical line represents the null estimate (with a value of 0 for pooled-ES). Horizontal lines that cross the null vertical line represent the non-significant overall summary of the meta-analysis.

**Figure 6 ijerph-18-10212-f006:**
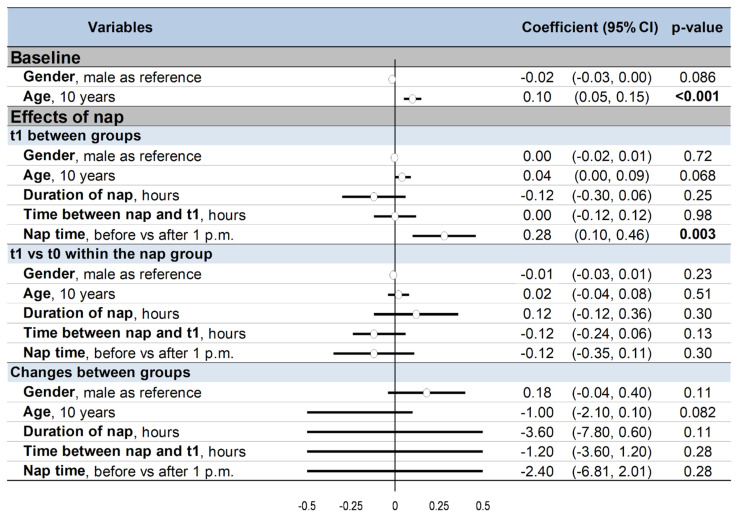
Summary of meta-regressions: 95%CI: 95% confidence intervals; t0: baseline; t1: after nap (nap group) or rest without nap (control group) Results were presented before (**left**) and after (**right**) exclusion of non-randomized controlled trials The effect of each variable on the outcome is represented in the forest-plot by a dot on a horizontal line. The black dots represent the coefficient for each variable, and the length of each line around the dots represents their 95% confidence interval (95CI). The black solid vertical line represents the null estimate (with a value of 0). Horizontal lines that cross the null vertical line represent non-significant variables on the outcome. Bold numbers represent the significant results (*p* < 0.05).

## Data Availability

All relevant data are within the paper.

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
