# Peer review of "Effects of a Short Daytime Nap on the Cognitive Performance: A Systematic Review and Meta-Analysis"

_ijerph, 2021, doi:10.3390/ijerph181910212_

Round 1

Reviewer 1 Report

Dear authors,

thank you very much for your paper. I would like to suggest putting some tables into the supplementary materials because of their length. This regards table 1 and figure 4.

Additionally, it would greatly benefit the paper, if you could spell check it with a native english speaker. 

I highlighted some sentences and words in the pdf.

Best wishes

Author Response

Reviewer 1

Dear authors, thank you very much for your paper. I would like to suggest putting some tables into the supplementary materials because of their length. This regards table 1 and figure 4. Additionally, it would greatly benefit the paper, if you could spell check it with a native english speaker. I highlighted some sentences and words in the pdf.

[REPLY] Thank you for your comment. As suggested, table 1 and figure 4 are now in supplementary materials. For Table 1, we are sorry – there was an error in its format that we did not see during the generation of the manuscript through the website. The correct length of Table 1 is only one page, in landscape mode, with readable data. As suggested, it is part of supplementary materials now, but we are happy to move it back within the manuscript if you prefer (in landscape mode). A native English speaker also proof-readed the article.

Reviewer 2 Report

  1. The design of table 1 isn't acceptable. Data aren't readable.
  2. One study with daytime naps in the usual work environment [41] should be explained in the limitation of the study because the environments of the sleep lab are completely different than social rooms for residents in hospitals.
  3. The activity of the control group during nap should be explained
  4. The nap definition is not clear, Is nap duration is the time of sleep duration or time to the possibility of sleep? It should be explained in detail.

Author Response

Reviewer 2

The design of table 1 isn't acceptable. Data aren't readable.

[REPLY] Thank you for your comment. Sorry, there was an error in the format of Table 1 that we did not see during the generation of the manuscript through the website. As suggested bu reviewer 1, Table 1 is now a Supplementary Table 1 (available in Supplementary online materials), and is now in landscape mode, with readable data.

One study with daytime naps in the usual work environment [41] should be explained in the limitation of the study because the environments of the sleep lab are completely different than social rooms for residents in hospitals.

[REPLY] Thank you for your comment. The Limitations section now reads: “Nearly all data were also in laboratory condition, except one study. Consequently, it is hard to make conclusions of effects of napping on cognitive performance during daytime work, decreasing the external validity[107] and strongly nuancing the level of evidence for napping at work. The environment differs widely between working conditions and sleep lab rooms in hospitals.”

The activity of the control group during nap should be explained

[REPLY] Thank you for your comment. The Results section now reads: “Activity of control group during nap: During the nap of the intervention group, the control group had the opportunity to rest[41,42,44,45,48,48,50] or to do a quiet activity like watching TV,[6,43,47] read[46] or play a card game.[45]”

The nap definition is not clear, Is nap duration is the time of sleep duration or time to the possibility of sleep? It should be explained in detail.

[REPLY] Thank you for your comment. The Results section now reads: “Duration of nap: Mean duration of nap i.e. time of sleep duration was 55.4±29.4 min, varying from 15[47] to 90 min.[6,44,45,50]”

Reviewer 3 Report

This is a well-executed review and meta-analysis on daytime napping and cognitive function. The findings are not extremely surprising, but it is clear that the authors were meticulous in their methodology and did a tremendous amount of work to synthesize the literature. I believe some of the tables and figures could be combined and/or moved to supplemental for ease of reading. There are grammatical errors throughout and the manuscript would benefit from a native English speaker's edits.

Specific comments:

  1. Abstract: The authors should reconsider softening the emphasis on napping at work, as most of the data in their review is from the laboratory.
  2.  Abstract/Highlights/Intro: The same phrase, which is also grammatically incorrect, appears 3x : "Napping in the workplace is occupying an increasing place". Could be changed to "Napping in the workplace is increasing.." or something similar, but is redundant when read 3x in a row.
  3. Figure 2. The risk of bias summary (2nd panel) is redundant with the above information. I suggest either removing the second panel, or adding a % green, %red, % yellow, Total% rows to the bottom of the first panel. With the color coding on the first panel, it is fairly easy for the reader to see the risk in each column.
  4. Results: The conclusion of the sentence from line 261-266 is hard to follow. It is unclear what "and demonstrated similar results except for executive functions" means in this case.
  5. Consider changing the category "after exclusion from metafunnel" to "Outliers excluded" for purposes of clarity. 
  6. Figure 4 demonstrates the thorough-ness of the analyses, but would probably be best in the Supplemental, given the number of figures and tables already in the text and the density of the Table.
  7. It is not clear what the asterisks mean in Figures 5 and 6
  8. The term "changes between groups" is confusing (Figure 6). Based on Section 3.9, my understanding is that this represents the comparison of the two groups in the study (control versus baseline). If so, it is not the change between groups, but rather the difference between them. Whereas both the "nap group" column and the "control group" column do actually represent a change in the score of one group across timepoint, if I understand correctly. If this is correct, simply omitting the word "changes" would clarify. If not, some further explanation of what it is (changes in napping group minus changes in control?) would also help.
  9. Discussion, section 4.2. In discussion of the "2 hrs following awakening", I would clarify that the findings are for 30-120 min after waking. I appreciate it is somewhat difficult to discuss given that findings at 30 min were significant until exclusion of outliers, so it is somewhat ambiguous. However, given the focus on discussing sleep inertia, I would take care to keep the language around this (within 2 h vs 30-120 min) consistent when possible. 
  10. Some reference should be made to the fact that clock time was used, rather than circadian (internal) time, especially since the authors raise circadian rhythms in alertness/sleepiness as a discussion point.
  11. The statement (lines 416-7) that the laboratory "conditions were as closest possible to real work" seems unfounded, as most studies were in the laboratory.

Author Response

Reviewer 3

This is a well-executed review and meta-analysis on daytime napping and cognitive function. The findings are not extremely surprising, but it is clear that the authors were meticulous in their methodology and did a tremendous amount of work to synthesize the literature.

[REPLY] Thank you for your very positive comment. Much appreciated.

I believe some of the tables and figures could be combined and/or moved to supplemental for ease of reading.

[REPLY] Thank you for your comment. As you rightly suggested and as reviewer 1 also suggested, Table 1 and Figure 4 are now within supplementary materials. List of Figures has been edited accordingly throughout the manuscript.

There are grammatical errors throughout and the manuscript would benefit from a native English speaker's edits.

[REPLY] Thank you for your comment. A native English speaker has proof-read the article.

Specific comments:

Abstract: The authors should reconsider softening the emphasis on napping at work, as most of the data in their review is from the laboratory.

 [REPLY] Thank you for your comment. The aim of the abstract now reads: “In this systematic review and meta-analysis, we aimed to assess the benefits of a short daytime nap on cognitive performance.” The conclusion of the abstract now reads: “Despite our meta-analyses included almost exclusively laboratory studies, daytime napping in the afternoon improved cognitive performance with benefits effects of early nap. More studies in real work condition are warranted before implementing daytime napping at work as a preventive measure to improve work efficiency.” We also updated the conclusion of the manuscript that now reads: “However, our meta-analyses included almost exclusively laboratory studies. Before recommending daytime napping at work as a preventive strategy, further studies should evaluate effects of nap on cognitive performance in real work condition to make the results more generalizable.” We also updated the highlights accordingly.

Abstract/Highlights/Intro: The same phrase, which is also grammatically incorrect, appears 3x : "Napping in the workplace is occupying an increasing place". Could be changed to "Napping in the workplace is increasing." or something similar, but is redundant when read 3x in a row.

[REPLY] Thank you for your comment. The sentence in the Abstract now reads: “Napping in the workplace is under debate, with interesting results on work efficiency and well-being of workers.” The sentence in the Highlights now reads: “Napping in the workplace may benefit to work efficiency and well-being of workers.” The sentence in the Introduction now reads: “Napping in the workplace is under consideration, with putative benefits on work efficiency and well-being of workers.”

Figure 2. The risk of bias summary (2nd panel) is redundant with the above information. I suggest either removing the second panel, or adding a % green, %red, % yellow, Total% rows to the bottom of the first panel. With the color coding on the first panel, it is fairly easy for the reader to see the risk in each column.

[REPLY] Thank you for your comment. We removed the second panel. We totally agree that it is better now. Thanks.

Results: The conclusion of the sentence from line 261-266 is hard to follow. It is unclear what "and demonstrated similar results except for executive functions" means in this case.

[REPLY] Thank you for your comment. The section now reads: “Overall cognitive performance at baseline (t0) did not differ between nap vs control groups (effect size -0.03, 95% CI -0.14 to 0.07, I2=21.2%), nor after exclusion of outliers [41,44,50] (-0.01, -0.10 to 0.09, I2=0.0%), nor after exclusion of non-RCT [41,42,47–49] (-0.04, -0.16 to 0.09, I2=17.2%) (Figure S1 and S2). Stratification by cognitive functions (memory, alertness, and executive functions) demonstrated similar results i.e. no difference between nap and control groups for the overall analyses, as well as after exclusion of outliers [41,44,50] or non-RCT[41,42,47–49] – except for executive functions that were lower in the nap group compared to the control group but only in the sensitivity analyses with two RCT (Figure S1 and S2).”

Consider changing the category "after exclusion from metafunnel" to "Outliers excluded" for purposes of clarity.

[REPLY] Thank you for your comment. Amended throughout the manuscript.

Figure 4 demonstrates the thoroughness of the analyses, but would probably be best in the Supplemental, given the number of figures and tables already in the text and the density of the Table.

[REPLY] Thank you for your comment. As you rightly suggested and as reviewer 1 also suggested, Table 1 and Figure 4 are now within supplementary materials. List of Figures has been edited accordingly throughout the manuscript.

It is not clear what the asterisks mean in Figures 5 and 6

[REPLY] Thank you for your comment. For clarity, we removed the asterisks to “After exclusion*” and we wrote “After exclusion of outliers” in Figures 5 and 6 (that became Figures 4 and 5 because Figure 4 is now within Supplementary materials. We also updated accordingly (removing the asterisks) all the figures within Supplementary materials.

The term "changes between groups" is confusing (Figure 6). Based on Section 3.9, my understanding is that this represents the comparison of the two groups in the study (control versus baseline). If so, it is not the change between groups, but rather the difference between them. Whereas both the "nap group" column and the "control group" column do actually represent a change in the score of one group across timepoint, if I understand correctly. If this is correct, simply omitting the word "changes" would clarify. If not, some further explanation of what it is (changes in napping group minus changes in control?) would also help.

[REPLY] Thank you for your comment. We amended the titles of the columns for an easier comprehension. The titles now reads: “Changes (t1-t0) within nap group”, “Changes (t1-t0) within control group” and “Changes (t1-t0) between groups”. The legend of the figure reads: “meta-analyses on cognitive performance at t1 compared with baseline (t0) within the nap group (left), meta-analyses on cognitive performance at t1 compared with baseline (t0) within the control group (middle), meta-analyses on changes in performance between t1 and t0 ((t1-t0)/t0) between nap and control groups (right).”

Discussion, section 4.2. In discussion of the "2 hrs following awakening", I would clarify that the findings are for 30-120 min after waking. I appreciate it is somewhat difficult to discuss given that findings at 30 min were significant until exclusion of outliers, so it is somewhat ambiguous. However, given the focus on discussing sleep inertia, I would take care to keep the language around this (within 2 h vs 30-120 min) consistent when possible.

[REPLY] Thank you for your comment. We totally agree with you. We updated accordingly.

Some reference should be made to the fact that clock time was used, rather than circadian (internal) time, especially since the authors raise circadian rhythms in alertness/sleepiness as a discussion point.

[REPLY] Thank you for your comment. The discussion now reads: “Our organism is physiologically programmed to rest in the afternoon. Our biological clock controls a biphasic rhythm with two periods conducive to sleep, varying with body temperature. The first peak of drowsiness occurs between 1 and 5 a.m., the second twelve hours later, i.e. between 1 to 5 p.m. [90]. The decrease in alertness within the afternoon is wrongly associated with digestion [91], but is mainly due to our circadian rhythm [92,93]. So it could suggest a night sleep and a nap in the early afternoon i.e. when we are naturally less vigilant [90]. However, ideally, workers may also benefit to nap according to their circadian rhythm rather than clock time.”

The statement (lines 416-7) that the laboratory "conditions were as closest possible to real work" seems unfounded, as most studies were in the laboratory.

[REPLY] Thank you for your comment. The sentence now reads: “As our meta-analyses included mainly lab studies, applicability of our results to real work must be inferred [101].”